# Evaluation of the Protective Immune Response Induced by an *rfbG*-Deficient *Salmonella enterica* Serovar Enteritidis Strain as a Live Attenuated DIVA (Differentiation of Infected and Vaccinated Animals) Vaccine in Chickens

Xinwei Wang,[a,b,c,d] Xilong Kang,[b,c,d] Mingxing Pan,[b,c,d] Ming Wang,[a,b,c,d] Jiayue Zhang,[b,c,d] Hongqin Song[a,b,c,d]

[a]College of Veterinary Medicine, Yangzhou University, Yangzhou, Jiangsu, China

[b]Jiangsu Key Laboratory of Zoonosis, Jiangsu Co-Innovation Center for Prevention and Control of Important Animal Infectious Diseases and Zoonoses, Yangzhou University, Yangzhou, Jiangsu, China

[c]Key Laboratory of Prevention and Control of Biological Hazard Factors (Animal Origin) for Agri-food Safety and Quality, Ministry of Agriculture of China, Yangzhou University, Yangzhou, Jiangsu, China

[d]Joint International Research Laboratory of Agriculture and Agri-product Safety of the Ministry of Education, Yangzhou University, Yangzhou, Jiangsu, China

**ABSTRACT** *Salmonella enterica* serovar Enteritidis (*S.* Enteritidis), one of the zoonotic pathogens, not only results in significant financial losses for the global poultry industry but also has the potential to spread to humans through poultry and poultry products. Vaccination is an effective method to prevent *Salmonella* infections. In this study, we constructed a live attenuated DIVA (differentiation of infected and vaccinated animals) vaccine candidate, Z11Δ*rfbG*, and evaluated its protective effectiveness and DIVA potential in chickens. Compared to that of the virulent wild-type strain, the 50% lethal dose ($LD_{50}$) of the *rfbG* mutant strain increased 56-fold, confirming its attenuation. High serum levels of *S.* Enteritidis-specific IgG titers indicated that a significant humoral immune response was induced in the vaccinated group. After challenge, the nonvaccinated group showed serious clinical symptoms (diarrhea, depression, decreased appetite, ruffled feathers, and weight loss), pathological changes (white nodules in the liver and fatty lesions in liver cells), and death. In contrast, there were no clinical symptoms, pathological changes, or death in the $5 \times 10^6$- and $5 \times 10^7$-CFU-vaccinated groups. Z11Δ*rfbG* vaccination significantly reduced *S.* Enteritidis colonization in the spleen, liver, and cecum. In addition, the Z11Δ*rfbG*-vaccinated group exhibited a negative response to the serological test, whereas the virulent wild-type Z11 infection group was strongly positive for the serological test, showing a DIVA capability of Z11Δ*rfbG* vaccination. Overall, our findings demonstrate the viability of the *rfbG* mutant as a live attenuated chicken vaccine that can discriminate between animals that have been immunized and those that have been infected.

**IMPORTANCE** *S.* Enteritidis is a highly adapted pathogen that causes significant economic losses in the poultry industry around the world. Vaccination is an effective method of controlling *S.* Enteritidis infections. Here, we demonstrated that *S.* Enteritidis Z11Δ*rfbG* has the potential to be a safe, immunogenic, and DIVA vaccine candidate for the control of *Salmonella* infections in chickens. Z11Δ*rfbG* not only provided effective protection in chickens but also distinguished between infected and vaccinated chickens by serological tests.

**KEYWORDS** *Salmonella* Enteritidis, *rfbG*, live attenuated vaccine, DIVA, chicken

Address correspondence to Hongqin Song, hqsong@yzu.edu.cn.

The authors declare no conflict of interest.

S almonellosis is a broad term that refers to acute and chronic infections caused by *Salmonella* in animals and humans and is a serious public health concern (1). *Salmonella*, a Gram-negative rod, is a major foodborne zoonotic pathogen with a wide

**TABLE 1** LD$_{50}$ of *S*. Enteritidis Z11 and Z11Δ*rfbG* strains in chickens

| Strain and challenge dose (CFU) | No. of deaths/total no. of chickens | LD$_{50}$ (CFU) |
|---|---|---|
| Z11 | | $1.8 \times 10^8$ |
| $5 \times 10^9$ | 12/12 | |
| $5 \times 10^8$ | 9/12 | |
| $5 \times 10^7$ | 2/12 | |
| $5 \times 10^6$ | 0/12 | |
| $5 \times 10^5$ | 0/12 | |
| Z11Δ*rfbG* | | $1 \times 10^{10}$ |
| $5 \times 10^{10}$ | 12/12 | |
| $5 \times 10^9$ | 2/12 | |
| $5 \times 10^8$ | 0/12 | |
| $5 \times 10^7$ | 0/12 | |
| $5 \times 10^6$ | 0/12 | |
| Blank, PBS | 0/12 | |

host range (2). *Salmonella enterica* serovar Enteritidis (*S*. Enteritidis) is a global concern, causing recessive infections in adult chickens and bacterial excretion into the external environment through feces, resulting in challenging pathogen purification, serious systemic infections, and high mortality (3). In recent years, it has been reported that *S*. Enteritidis is gradually replacing *Salmonella enterica* serovar Typhimurium as the most common foodborne *Salmonella* serotype (4). Additionally, the widespread use of antibiotics has led to the emergence of multiple antibiotic-resistant bacteria. Therefore, there is an urgent need for an efficient vaccine to manage this major zoonotic infection (4). Owing to their ability to stimulate cellular and humoral adaptive immune responses, live *Salmonella* vaccines are considered more effective against intestinal and systemic infections than inactivated vaccines (5, 6).

Lipopolysaccharide (LPS), an important component of the outer membrane of Gram-negative rods, primarily influences the phenotype, virulence, and immune cross-reaction of *Salmonella* strains (7, 8). LPS is primarily composed of O antigen, core polysaccharides, and lipid A (9). The *rfa*, *rfb*, and *rfc* gene clusters are the main genes involved in LPS biosynthesis (10). The *rfb* gene cluster regulates biosynthesis of the O antigen chain, with *rfbG* encoding CDP-glucose 4,6-dehydratase dehydratase (11). Jiao et al. (12) showed that *rfbG* downregulation leads to LPS deficiency in *S*. Enteritidis, which reduces *S*. Enteritidis virulence in a mouse model, and that its gene locus can be used as a marker. However, the effects of the *S*. Enteritidis *rfbG*-deficient strain on chicken remain unknown.

This study aimed to determine whether attenuated levels of *S*. Enteritidis *rfbG*-deletion mutants in a chicken model would protect poultry against virulent *S*. Enteritidis challenge, while also allowing us to distinguish between the sera of vaccinated and infected chickens.

## RESULTS

**Virulence of the vaccine candidates.** The virulence of the Z11Δ*rfbG* and Z11 strains (*rfbG*-deletion mutant strain and wild-type *S*. Enteritidis strain, respectively) was evaluated in 7-day-old specific-pathogen-free (SPF) chickens after intramuscular immunization. As shown in Table 1, the 50% lethal dose (LD$_{50}$) of Z11Δ*rfbG* was $1 \times 10^{10}$ CFU, which was 56-fold higher than that of the virulent wild-type Z11 ($1.8 \times 10^8$ CFU). These results indicated that the virulence of Z11Δ*rfbG* was attenuated compared to that of the wild-type strain.

**Immune protective efficacy of Z11Δ*rfbG* after challenge.** To evaluate the protective efficacy of Z11Δ*rfbG*, chickens were vaccinated intramuscularly with Z11Δ*rfbG* and challenged with the virulent wild-type strain Z11. The survival percentages of these chickens are shown in Table 2. Following the challenge with Z11, two chickens died in

**TABLE 2** Protective efficacy of intramuscular Z11Δ*rfbG* vaccination

| Vaccination | | | | Challenge | | | No. of survivors/ | Survival |
|---|---|---|---|---|---|---|---|---|
| Strain | Route | Dose (CFU) | No. | Strain | Route | Dose (CFU) | no. total | rate (%) |
| Z11Δ*rfbG* | Intramuscularly | $5 \times 10^5$ | 15 | Z11 | Intramuscularly | $7 \times 10^8$ | 13/15 | 86.7 |
| | | $5 \times 10^6$ | | | | | 15/15 | 100 |
| | | $5 \times 10^7$ | | | | | 15/15 | 100 |
| PBS | | | | | | | 11/15 | 73.3 |
| PBS | | | | | | | 15/15 | 100 |

the $5 \times 10^5$-CFU-vaccinated group. Meanwhile, 4 out of 15 nonvaccinated chickens died in the control group. However, none of the chickens in the $5 \times 10^6$- and $5 \times 10^7$-CFU-vaccinated groups died. Slight and temporary diarrhea was observed following challenge in the vaccinated group compared to the control group, whereas persistent weight loss, diarrhea, decreased appetite, ruffled feathers, and depression were observed following challenge in the nonvaccinated group.

**Clearance of bacteria in the spleen, liver, and cecum.** Before challenge (7 and 10 days after the second immunization), no bacteria were isolated from the liver, spleen, and cecum of the vaccinated group and the nonvaccinated group, indicating that there were no *rfbG* deletion strains in the vaccinated group and the nonvaccinated group. After challenge with Z11, the bacteriological analysis of chicken organs showed that the bacterial load of all liver, spleen, and cecum samples in the control group was negative. Figure 1A shows the dynamic changes of bacteria in the liver after challenge. On the 5th day after challenge, the bacterial load of the $5 \times 10^7$-CFU-vaccinated group was completely cleared, and the bacterial load of the $5 \times 10^6$-CFU-vaccinated group was partially cleared, but the bacterial load of the nonvaccinated group was maintained at a high level. Figure 1B shows the dynamic changes of bacteria in the spleen after challenge. Within 1 to 14 days postchallenge (DPC), the bacterial load of the vaccinated group was lower than that of the nonvaccinated group. On the 14th day after challenge, the bacterial load of the vaccinated group was partially cleared, while that of the nonvaccinated group was still high. Figure 1C shows the dynamic changes of bacteria in the cecum after challenge. On the 14th day after challenge, the bacterial load in the vaccinated group was partially cleared, while that in the nonvaccinated group was still high. These results indicated that the vaccine promoted bacterial clearance from the liver, spleen, or cecum.

**Body weight after challenge.** The changes of the body weight of chickens after challenge were monitored. As shown in Fig. 2, the body weight of both the vaccinated group and the nonvaccinated group decreased within 1 to 3 days after challenge. With the clearance of bacteria, the body weight of the vaccinated group began to rise at 5 DPC, while the body weight of the nonvaccinated group still decreased on the 5th to 9th day and reached its lowest value at 9 DPC.

**Histological analysis after challenge with Z11.** Liver lesions were observed at 14 DPC, and histological analysis of the liver was performed using hematoxylin and eosin (H&E) staining. As shown in Fig. 3, no obvious lesions were detected in the livers of the $5 \times 10^6$- and $5 \times 10^7$-CFU-vaccinated groups compared to those of the control group. However, white nodules were observed in the liver of the nonvaccinated group (Fig. 3A), and the liver cells of the $5 \times 10^5$-CFU-vaccinated group and the nonvaccinated group had fatty changes which were more severe in the nonvaccinated group (Fig. 3B).

**Humoral immune responses after immunization.** Serum *S.* Enteritidis-specific IgG antibodies were evaluated in chickens following Z11Δ*rfbG* immunization to determine the efficacy of Z11Δ*rfbG* in inducing humoral immune responses. As shown in Fig. 4, IgG antibodies against *S.* Enteritidis in the Z11Δ*rfbG*-vaccinated group were detected and increased dramatically 10 days after the second immunization. The $5 \times 10^6$- and

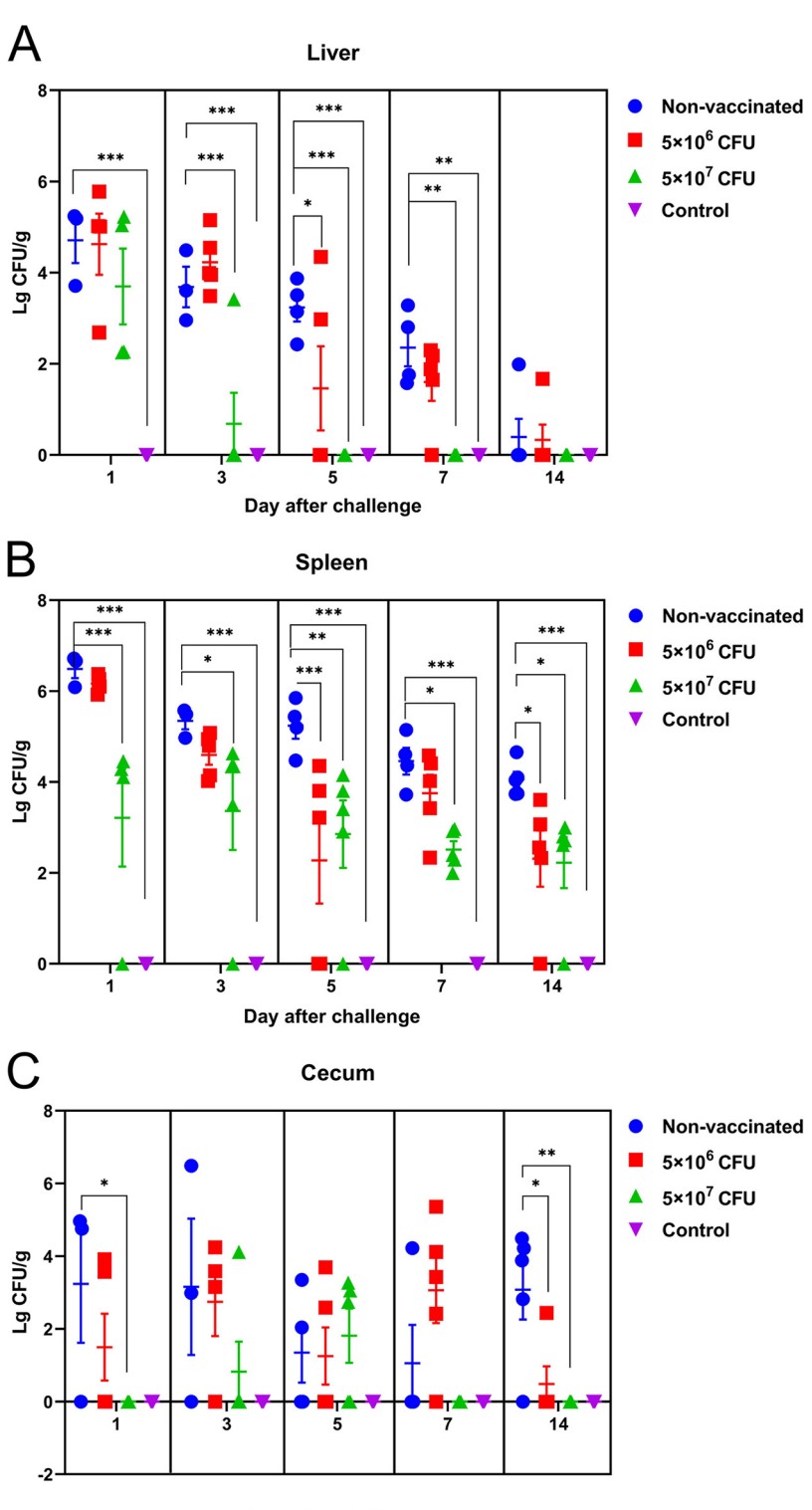

**FIG 1** Colonization of *Salmonella* Enteritidis in chicken organs after challenge. Colonization and persistence of the same dose of Z11 in liver (A), spleen (B), and cecum (C) of vaccinated and nonvaccinated chickens. The control group received 100 $\mu$L of PBS. *, $P < 0.05$; **, $P < 0.01$; and ***, $P < 0.001$, compared with the bacterial load of nonvaccinated group chickens by one-way ANOVA followed by Bonferroni's multiple-comparison test. Data are presented as mean $\pm$ SEM in $\log_{10}$ CFU per gram.

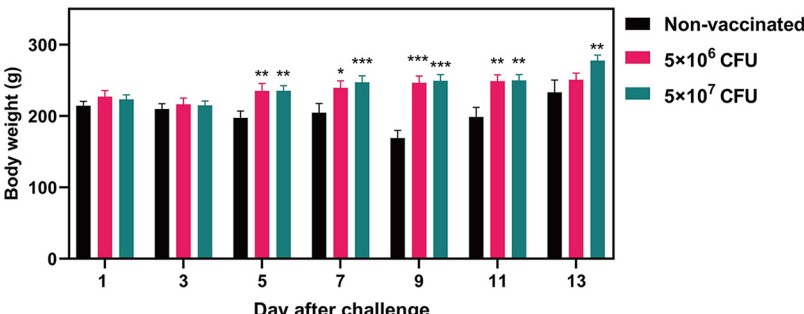

**FIG 2** Body weight of chickens after the challenge. Chickens from vaccinated and nonvaccinated groups were intramuscularly challenged with $1 \times 10^9$ CFU of the virulent wild-type strain (Z11). The body weights of these chickens were determined at 1, 3, 5, 7, 9, 11, and 13 DPC. *, $P < 0.05$; **, $P < 0.01$; and ***, $P < 0.001$, compared with the bodyweight of nonvaccinated chickens by one-way ANOVA followed by Bonferroni's multiple-comparison test. Data are presented as mean $\pm$ SEM.

$5 \times 10^7$-CFU-vaccinated groups had significantly higher *S*. Enteritidis-specific IgG antibody titers than the control group.

**The DIVA capability of the Z11Δ*rfbG* vaccine.** The DIVA (differentiation of infected and vaccinated animals) capability of the Z11Δ*rfbG* vaccine was evaluated using the slide agglutination test or the BioCheck *Salmonella* group D antibody enzyme-linked immunosorbent assay (ELISA) to detect LPS-specific serum antibodies. The slide agglutination test showed that the serum samples from chickens immunized with Z11Δ*rfbG* failed to agglutinate with the commercialized agglutination antigens on day 14 postimmunization (Fig. 5). However, serum samples from chickens infected with the virulent wild-type strain Z11 agglutinated with *S*. Enteritidis antigens (Fig. 5).

## DISCUSSION

*Salmonella enterica* causes high morbidity and death in humans, in terms of both the number of infections and the severity of the disease. In contrast to the *Salmonella enterica* serovars with a limited host range, such as *S*. Typhi, *S*. Dublin, and *S*. Gallinarum, *S*. Enteritidis has a broad host range (13). Additionally, poultry and poultry-associated products are among the most important vehicles for human *Salmonella* infections (14). Vaccination in chickens is an important strategy currently used to reduce the levels of *Salmonella* in poultry flocks (15). A live attenuated *Salmonella* vaccine must be sufficiently attenuated, immunogenic, and protective. A variety of *Salmonella* mutant strains have been suggested as candidate vaccines. In a previous study, we constructed a mutant of the *S*. Enteritidis Z11 strain that lacks *rfbG* (16). The objective of the present study was to evaluate the safety, protective efficacy, and DIVA capability of the *rfbG*-deficient mutant *S*. Enteritidis Z11 strain as a live attenuated DIVA vaccine candidate.

Live attenuated *Salmonella* vaccines are avirulent. Some LPS-deficient mutants result in attenuation of *Salmonella* (17). Jiao et al. (12) showed that the virulence of *rfbG* mutant strains was attenuated in a mouse model and that *rfbG* mutant strains are safe for mammals. Similarly, this study showed that the $LD_{50}$ of Z11Δ*rfbG* was 56-fold higher than that of wild-type Z11 in a chicken model, indicating that the virulence of Z11Δ*rfbG* was significantly attenuated compared to that of the virulent wild-type strain Z11 in chickens. In contrast, *rfaH*, an essential gene for LPS biosynthesis, does not affect the virulence of *S*. Gallinarum (18). Although the virulence of *Salmonella enterica* serovar Pullorum (*S*. Pullorum) S06004Δ*spiC*Δ*rfaH* was significantly attenuated compared to that of wild-type S06004, it was mainly the *spiC* deletion that exerted the attenuating effect (19).

The early immune response against *Salmonella* relies on innate immunity within the gut mucosa. In general, antibodies are necessary for resistance to systemic *Salmonella* infections, and as the infection progresses, an effective immune response to *Salmonella*

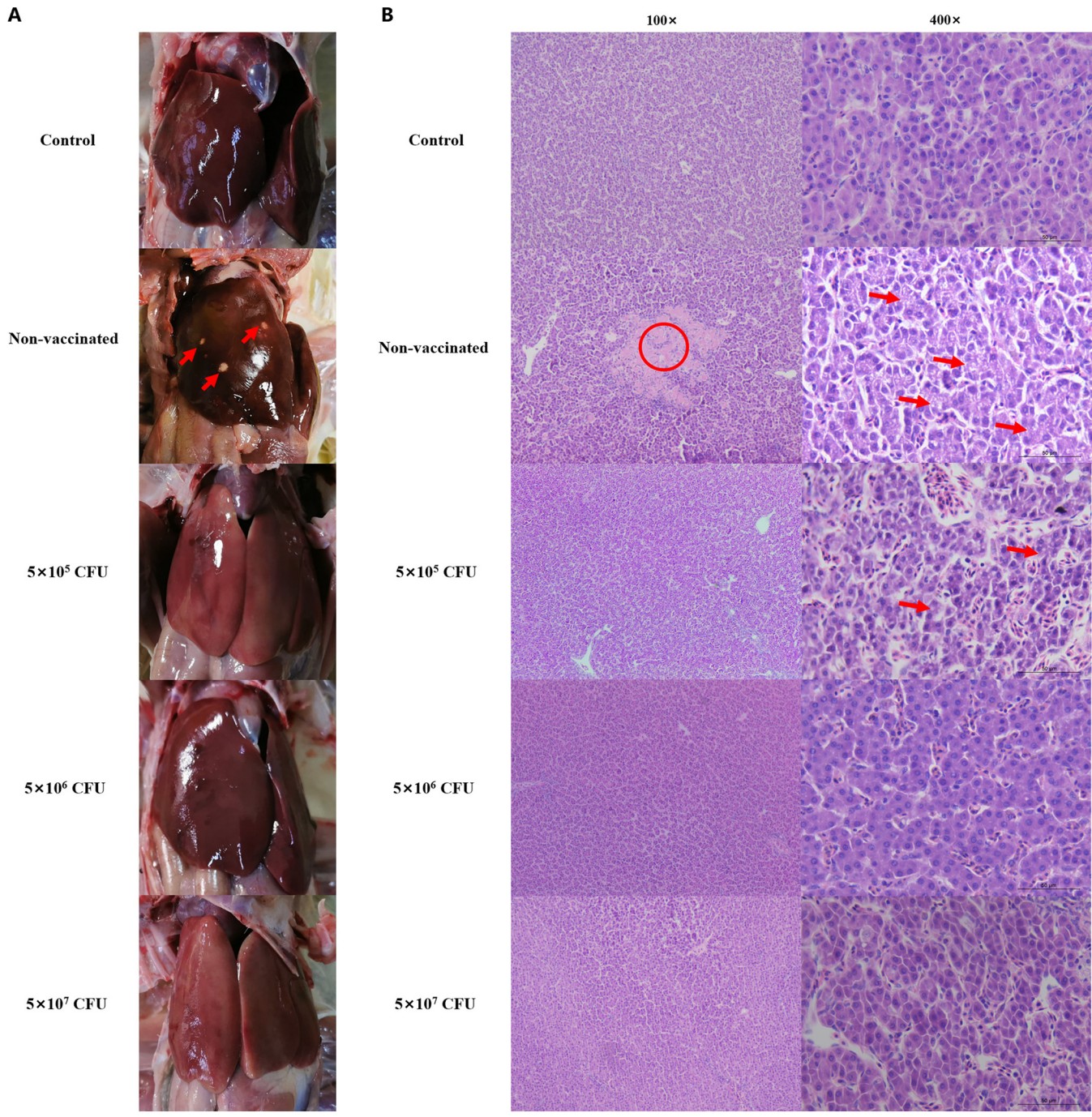

**FIG 3** Images showing the pathological anatomy (A) and histological analysis (B) of the liver after the bacterial challenge. (A) The liver pathological anatomy was observed 14 days postchallenge (DPC). The arrows indicate the lesions. (B) The histopathological changes in the livers of chickens were examined by H&E staining after 14 days of challenge. The results were observed at ×100 and ×400 magnification using an optical microscope. Arrows in the liver sections represent the fatty changes. The circle in the liver section indicates the necrotic foci.

relies on humoral immunity, which can provide effective protection to the host (20, 21). A previous study reported that a novel live *S*. Enteritidis vaccine strain, JOL919, induced significant systemic immunoglobulin responses in chickens after inoculation (22). Another study reported that IgG levels were significantly higher in chickens vaccinated with the *cobS* and *cbiA* mutant *S*. Gallinarum strains than in the control group (23). Similarly, in this study, all chickens immunized with $5 \times 10^6$ and $5 \times 10^7$ CFU of Z11Δ*rfbG* had significantly higher levels of *S*. Enteritidis-specific serum IgG than the nonvaccinated group. These results demonstrate that the *rfbG* mutant can induce strong humoral responses.

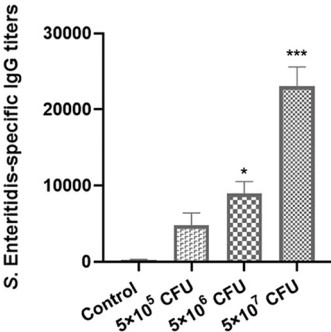

**Day 10 after the second immunization**

**FIG 4** Determination of serum IgG levels. An enzyme-linked immunosorbent assay was used to identify *S.* Enteritidis-specific IgG antibody titers in the serum of chickens from each group 10 days following the second inoculation. *, $P < 0.05$; and ***, $P < 0.001$, compared with the control group by one-way ANOVA followed by Bonferroni's multiple-comparison test. Data are presented as mean ± SEM.

A standard vaccine not only induces a strong immune response but also has high immune protection efficacy. In this study, the survival rate of the $5 \times 10^7$- and $5 \times 10^6$-CFU-vaccinated groups was 100% after two intramuscular injections. Furthermore, chickens immunized with $5 \times 10^6$ and $5 \times 10^7$ CFU of Z11Δ*rfbG* showed no clinical symptoms. Slight pathological changes were observed in the organs of $5 \times 10^5$-CFU-vaccinated chickens, whereas more severe fatty changes were detected in unvaccinated chickens after the challenge. *S.* Enteritidis colonizes the gastrointestinal tract in avians, which is frequently followed by an invasion of systemic areas, including the spleen and liver (24). Braukmann et al. (25) showed that a live *Salmonella enterica* vaccine effectively inhibited the invasion and colonization of the challenge strain. Furthermore, a study reported that bacterial clearance occurred 1 week after *Salmonella* infections (26). Springer et al. (27) demonstrated that a live *Salmonella* Enteritidis vaccine inhibits systemic invasion after early infection with *S.* Typhimurium and *Salmonella enterica* serovar Infantis (*S.* Infantis). Our results showed that intramuscular immunization with *S.* Enteritidis Z11Δ*rfbG* in chickens accelerated the clearance of the virulent wild-type Z11 in the liver, spleen, and cecum after challenge, which is consistent with previous studies (25–27). Therefore, the *rfbG* mutant showed a strong protective efficacy.

Although vaccination may interfere with established serological testing procedures and make it difficult to distinguish between immunized and infected animals, with the development of vaccine technology, DIVA vaccine can be used to distinguish between immunized and infected animals. Most of the attenuated mutant strains used in DIVA vaccines lack specific antigens (such as LPS and neuraminidase); therefore, LPS is a suitable target for the construction of the DIVA vaccine (28). We have previously shown that *rfbG* plays a role in LPS synthesis (12). In the present study, the Z11Δ*rfbG*-vaccinated group exhibited a negative serological test result, while the virulent wild-type Z11 infection group showed a strong positive result for the serological test. These



**FIG 5** DIVA capability of Z11Δ*rfbG*. The serum was collected from chickens infected with Z11Δ*rfbG*, Z11, or PBS for 14 days and used for the detection of LPS antibodies. An agglutination assay was performed using commercialized agglutination antigens.

results indicate that the *rfbG* mutant allows for the differentiation between infected animals and vaccinated animals. Therefore, it may be used in combination with herd-level *Salmonella* surveillance.

In conclusion, this study demonstrated that *S.* Enteritidis Z11Δ*rfbG* has the potential to be a safe, immunogenic, and DIVA vaccine candidate for the control of *Salmonella* infections. Attenuated *S.* Enteritidis Z11Δ*rfbG* elicited a strong humoral immune response and provided effective protection in chickens. In addition, vaccination with Z11Δ*rfbG* did not affect our ability to distinguish between infected and vaccinated chickens by serological tests.

## MATERIALS AND METHODS

**Experimental animals.** Healthy SPF chickens (7 days old) were obtained from Zhejiang Lihua Agricultural Technology Co., Ltd., China. All animal experiments were approved by the Animal Welfare and Ethics Committees of Yangzhou University and complied with the guidelines of the Institutional Administrative Committee and Ethics Committee of Laboratory Animals (IACUC license number: SYXK [Su] 2016-0020).

**Bacterial strains.** Virulent wild-type *S.* Enteritidis strain Z11 was a clinical isolate obtained from *S.* Enteritidis-infected chickens and stored in our laboratory. The *rfbG*-deletion mutant strain, Z11Δ*rfbG*, was constructed using a homologous recombination technique mediated by a suicide plasmid, as previously described (16). Briefly, upstream and downstream fragments of the *rfbG* gene were amplified using PCR. The pDM4 plasmid was digested using the restriction endonuclease XbaI (TaKaRa). The purified plasmid and the upstream and downstream fragments were fused using the ClonExpress MultiS one-step cloning kit (Vazyme Biotechnology Co., Ltd., Nanjing, Jiangsu, China). Recombinant plasmids were transferred into X7213 cells and sequenced. Single crossover mutants were obtained by conjugal transfer of recombinant suicide plasmids into the Z11 strain. The *rfbG*-deletion mutant was screened on 15% sucrose Luria-Bertani (LB) plates. We used PCR to verify whether *rfbG* is missing by amplifying the sequence with the outer primers and the inner primers of *rfbG*. The *S.* Enteritidis Z11Δ*rfbG* strain was used as a DIVA vaccine candidate in this study.

**Assessment of bacterial virulence.** The virulence of *S.* Enteritidis Z11 and Z11Δ*rfbG* vaccines was evaluated in chickens by determining the 50% lethal dose ($LD_{50}$). One hundred thirty-two 7-day-old SPF chickens were used in this study. Sixty chickens in the wild-type group were randomly assigned to five groups ($n = 12$). Each group was injected intramuscularly with a 10-fold dilution ($5 \times 10^5$ to $5 \times 10^9$ CFU) of Z11. Sixty chickens in the deletion mutant group were randomly assigned to five groups ($n = 12$). Each group was injected intramuscularly with a 10-fold dilution ($5 \times 10^6$ to $5 \times 10^{10}$ CFU) of Z11Δ*rfbG*. Twelve chickens were inoculated with 100 $\mu$L of phosphate-buffered saline (PBS) via the same route as the control group. Chicken death was monitored daily for 14 days postinfection. $LD_{50}$ was calculated using the Reed-Muench method (29).

**Immune protection assessment.** Seventy-five 7-day-old SPF chickens were randomly assigned to three groups, namely, the control group (no vaccine, no challenge) ($n = 15$), nonvaccinated group (no vaccine but challenge) ($n = 15$), and the vaccinated group (vaccine and challenge) ($n = 45$). SPF chickens in the vaccinated group were randomly assigned to three groups ($n = 15$). Each group was administered 100 $\mu$L of a diluted suspension of Z11Δ*rfbG* containing $5 \times 10^8$, $5 \times 10^7$, or $5 \times 10^6$ CFU/mL in PBS by intramuscular injection. Chickens in the control and nonvaccinated groups received 100 $\mu$L PBS via the same route. After the first immunization, the vaccine groups were administered a booster dose at 17 days of age. These chickens, as well as those in the unvaccinated group (28 days old), were challenged intramuscularly with $7 \times 10^8$ CFU of the Z11 strain 11 days after the second immunization. Deaths and clinical symptoms were recorded daily for 14 days after the challenge.

**Bacterial clearance assay.** Bacterial clearance in the internal organs of the chickens was evaluated. One hundred thirty 7-day-old SPF chickens were randomly assigned to three groups, namely, the control group (no vaccine, no challenge) ($n = 25$), nonvaccinated group (no vaccine but challenge) ($n = 35$), and the vaccinated group (vaccine and challenge) ($n = 70$). SPF chickens in the vaccinated group were randomly assigned to two groups ($n = 35$). Each group was administered 100 $\mu$L of a diluted suspension of Z11Δ*rfbG* containing $5 \times 10^8$ or $5 \times 10^7$ CFU/mL in PBS by intramuscular injection. Chickens in the control and nonvaccinated groups received 100 $\mu$L PBS via the same route. After the first immunization, the vaccine groups were administered a booster dose at 17 days of age. These chickens, as well as those in the unvaccinated group (28 days old), were challenged intramuscularly with $1 \times 10^9$ CFU of the Z11 strain 11 days after the second immunization. We carried out bacteriological analysis according to the description of Barrow and Lovell (30). Liver, spleen, and cecum samples of each group were collected aseptically on 1, 3, 5, 7, and 14 DPC. The samples were then weighed and homogenized in 1 mL of PBS. Homogenates of liver and spleen were serially diluted 10-fold and inoculated on LB agar plates, and homogenates of cecum were serially diluted 10-fold and inoculated on XLT4 plates and incubated at 37°C for 12 to 16 h. Bacterial colonies were calculated as $log_{10}$ CFU per gram.

**Changes in body weight after challenge.** The body weight of the vaccinated group and nonvaccinated group was recorded at 1, 3, 5, 7, 9, 11, and 13 DPC, respectively.

**Histological analysis.** Sections of the spleen and liver were collected from chickens at 14 DPC, and tissue samples were fixed in 10% neutral buffered formalin. Paraffin-embedded sections were stained with H&E (31) and were observed at ×100 and ×400 magnifications using an optical microscope.

**Serum IgG test.** Humoral immune responses were evaluated by determining the *S.* Enteritidis-specific IgG antibody levels using enzyme-linked immunosorbent assay (ELISA), as previously described (32), using Z11Δ*rfbG* as the coating antigen. Serum samples were collected from chickens in each group on day 10 after the second immunization and then serially diluted to be used as the primary antibody. The secondary antibody used was horseradish peroxidase (HRP)-conjugated rabbit anti-chicken IgG (1:10,000 dilution; Sigma-Aldrich, St. Louis, MO, USA). HRP activity was determined using a 3,30,5,50-tetramethylbenzidine substrate solution (Solarbio, Beijing, China), and the optical density at 450 nm ($OD_{450}$) was determined using an ELISA reader (BioTek, Winooski, VT, USA).

**DIVA capability assessment for the Z11Δ*rfbG* vaccine.** The DIVA capability of the Z11Δ*rfbG* strain was evaluated using the serological method to detect LPS-specific serum antibodies by a slide agglutination test or a commercial ELISA kit. Fifteen chickens were randomly divided into 3 groups ($n = 5$). Cells were infected with Z11Δ*rfbG*, Z11, or PBS. Serum was collected 14 days later and used to detect LPS antibodies. The slide agglutination test was performed using commercialized agglutination antigens obtained from Zhonghai Biotech Co., Ltd. (Beijing, China), according to the manufacturer's instructions. ELISA was performed using the *Salmonella* group D antibody test kit (BioCheck, Inc., San Francisco, CA, USA) according to the manufacturer's instructions (19).

**Statistical analysis.** GraphPad Prism 5 software (San Diego, CA, USA) was used for data analysis. The one-way analysis of variance (ANOVA) and Bonferroni's multiple-comparison test were employed to identify the significant differences between the experimental groups. Statistical significance was set as a $P$ value of $<0.05$ (\*), $<0.01$ (\*\*), or 0.001 (\*\*\*). Data are presented as mean $\pm$ standard error of the mean (SEM).

## ACKNOWLEDGMENTS

This work was funded by the National Natural Science Foundation of China (31972685, 32161143011, and 31902278), the China Postdoctoral Science Foundation (2018M642333), Jiangsu Province Policy Guidance Program (International Science and Technology Cooperation) (BZ2020013), the Research and Development Program of Jiangsu (BE2021354), and the Priority Academic Program Development of Jiangsu Higher Education Institutions (PAPD).

We gratefully acknowledge the help provided by Yi Zhou, Xia Huang, Yueyue Shang, Shunzi Han, and Ang Li in the animal experiments.

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
