## [Reviewer comments · Microbiology Spectrum]

Microbiology Spectrum

Evaluation of the protective immune response induced by an *rfbG*-deficient *Salmonella* Enteritidis strain as a live attenuated DIVA vaccine in chickens

Xinwei Wang, Xilong Kang, Mingxing Pan, Ming Wang, Jiayue Zhang, and Hongqin Song

Corresponding Author(s): Hongqin Song, Yangzhou University

Review Timeline:

Submission Date:	May 1, 2022
Editorial Decision:	June 7, 2022
Revision Received:	September 14, 2022
Accepted:	October 22, 2022

Editor: Mariola Edelmann

Reviewer(s): Disclosure of reviewer identity is with reference to reviewer comments included in decision letter(s). The following individuals involved in review of your submission have agreed to reveal their identity: Lisa Emerson (Reviewer #1); Wolfgang Koester (Reviewer #2)

Transaction Report:

DOI: <https://doi.org/10.1128/spectrum.01574-22>

June 7, 2022

Dr. Hongqin Song
Yangzhou University
Yangzhou
China

Re: Spectrum01574-22 (Evaluation of the protective immune response induced by an *rfbG*-deficient *Salmonella* Enteritidis strain as a live attenuated DIVA vaccine in chickens)

Dear Dr. Hongqin Song:

Link Not Available

Sincerely,

Mariola Edelmann

Journals Department
Reviewer comments:

Reviewer #1 (Comments for the Author):

Overall this is an interesting study with well supported conclusions. Authors should add all statistical test methods used in the paper in the methods section and figure legends. Additionally, authors can elaborate on the importance of being able to differentiate between vaccinated and infected chickens using sera.

Reviewer #2 (Comments for the Author):

Evaluation of the protective immune response induced by an *rfbG*-deficient *Salmonella* Enteritidis strain as a live attenuated

The authors claim to have constructed a rfbG mutant of Salmonella enterica serovar Enteritidis (SE) that would be suitable as a live attenuated DIVA vaccine strain with protective properties against SE infections.

The topic of SE as a pathogen for poultry, but more importantly as a zoonotic pathogenic bacterium affecting humans, thus being a public health concern, is timely. This reviewer can imagine how much effort was put into the study. Nonetheless, the manuscript appears at this state somewhat preliminary as all sections of the manuscript need to be more elaborated. Many sections appear rather cryptic and are lacking details that are essential for the understanding of the executed experiments. Grammar and style should be carefully checked.

A very important aspect, although mentioned in the discussion section, is totally excluded from their own study: the colonization of the gastrointestinal tract of the chickens after oral uptake/inoculation of SE. This plays an important role in the natural environment as well as in the small scale and large scale industrial settings of poultry production. Many issues of food poisoning through SE occur when contaminated poultry carcasses are processed. Protection of poultry against systemic infection is important, indeed. But what about protection against gastrointestinal colonization and shedding of SE into the environment? The main aspect still is the problem that this zoonotic pathogen can easily be transferred to humans via poultry products and hens eggs.

Specific comments:

- It is rather safe to say that reversion of the rfbG mutation can be excluded. But what about the possibility of recombination with DNA fragments from other Salmonella strains that could potentially be present in the same bird at the time of vaccination? Please discuss.
- Abstract; first sentence: other serovars can cause salmonellosis as well. Please re-phrase
- Line 33-36: Re-phrase, please. A lethal challenge is per definition deadly. How can it be that birds do not show any signs of disease after a lethal challenge?
- Line 42: Find another word for "transformed".
- Line 59 and throughout the document: "Gram" starts always with a capital "G".
- Complete results section: Please provide sufficient details in all sub-sections.
- Line 119 and later in manuscript: what are "fatty changes"? Please explain.
- Line 213: Do the authors mean "constructed" rather than "contrasted"?
- Line 221 -223: How was the rfbG mutation confirmed? By sequencing?
- Lines 246 ff: Which volume of the samples was plated onto LB plates? What is the detection limit? Samples should have undergone an enrichment procedure in addition to "direct plating" How would LB medium discriminate against bacterial strains other than the challenge strain?
- Line 253: Please explain H&E stain.
- Table 2: Please elaborate what is the difference between the two PBS groups?
- Table 1: The respective experiment should be executed in duplicate and repeated with a larger number of birds. Were the birds commingled or housed in different rooms (potential "pen effect") ?

Staff Comments:

Preparing Revision Guidelines

For complete guidelines on revision requirements, please see the journal Submission and Review Process requirements at <https://journals.asm.org/journal/Spectrum/submission-review-process>. **Submissions of a paper that does not conform to**

Microbiology Spectrum guidelines will delay acceptance of your manuscript. "

Please return the manuscript within 60 days; if you cannot complete the modification within this time period, please contact me. If you do not wish to modify the manuscript and prefer to submit it to another journal, please notify me of your decision immediately so that the manuscript may be formally withdrawn from consideration by Microbiology Spectrum.

Evaluation of the protective immune response induced by an *rfbG*-deficient *Salmonella*
Enteritidis strain as a live attenuated DIVA vaccine in chickens

Xinwei Wang,^{a,b,c,d} Xilong Kang,^{a,b,c,d} Mingxing Pan,^{a,b,c,d} Ming Wang,^{a,b,c,d} Hongqin
Song^{a,b,c,d}

6 ^a College of Veterinary Medicine, Yangzhou University, Yangzhou, Jiangsu 225009,
China

8 ^b Jiangsu Key Laboratory of Zoonosis, Jiangsu Co-Innovation Center for Prevention and
9 Control of Important Animal Infectious Diseases and Zoonoses, Yangzhou University,
Yangzhou, Jiangsu 225009, China

11 ^c Key Laboratory of Prevention and Control of Biological Hazard Factors (Animal
Origin) for Agri-food Safety and Quality, Ministry of Agriculture of China, Yangzhou
University, Yangzhou, Jiangsu 225009, China

14 ^d Joint International Research Laboratory of Agriculture and Agri-product Safety of the
15 Ministry of Education, Yangzhou University, Yangzhou, Jiangsu 225009, China

Running Head: *SALMONELLA* ENTERITIDIS DIVA VACCINE

#Address correspondence to Hongqin Song, hqsong@yzu.edu.cn.

*Present address: College of Veterinary Medicine, Yangzhou University, 48 East Wenhui
Road, Yangzhou, Jiangsu 225009, China

Word count of the abstract: 222

Word count of the text: 2536

**ABSTRACT:** Salmonellosis is a widespread zoonotic disease caused by *Salmonella*
*enterica* serovar Enteritidis (*S. Enteritidis*). *Salmonella* infections in humans are primarily
caused by poultry and poultry products. Vaccination is an effective method to prevent
*Salmonella* infections. In this study, we constructed a live attenuated DIVA
(differentiation of infected and vaccinated animals) vaccine candidate, Z11 Δ *rfbG*, and
evaluated its protective effectiveness and DIVA potential in chickens. Compared to the
virulent wild-type strain, the 50% lethal dose (LD₅₀) of the *rfbG* mutant strain increased
61-fold, confirming its attenuation. High serum levels of *S. Enteritidis*-specific IgG titers
indicated that a significant humoral immune response was induced in the vaccinated
group. The 5 \times 10⁶ and 5 \times 10⁷ colony-forming unit (CFU)-vaccinated chicken groups
showed no clinical symptoms, pathological changes, or death after the lethal challenge. In
contrast, the control group showed severe clinical symptoms and pathological alterations.
Z11 Δ *rfbG* vaccination significantly reduced *S. Enteritidis* colonization in the spleen and
liver. In addition, the Z11 Δ *rfbG* vaccinated group exhibited a negative response to the
serological test, whereas the virulent wild-type Z11 infection group was strongly positive
for the serological test, showing a DIVA capability of Z11 Δ *rfbG* vaccination. Overall, our
findings show that the *rfbG* mutant strain may be transformed into a live attenuated
vaccine that can be used in chickens without compromising the ability to differentiate
between the sera of infected and vaccinated animals.

**IMPORTANCE:** *S. Enteritidis* is a highly adapted pathogen that causes significant
economic losses in the poultry industry around the world. Vaccination is an effective
method of controlling *S. Enteritidis* infections. Here, we demonstrated that *S. Enteritidis*

redacted
2022-05-26 18:14:12

Line 43: Please indicate why
differentiating
between infected and vaccinated
animals is important
significant
is an effective

Z11 $\Delta rfbG$ has the potential to be a safe, immunogenic, and DIVA vaccine candidate for
the control of *Salmonella* infections in chickens. Z11 $\Delta rfbG$ not only provided effective
protection in chickens, but also had ability to distinguish between infected and vaccinated
chickens by serological tests.

**Keywords:** *Salmonella* Enteritidis, *rfbG*, live attenuated vaccine, DIVA, Chicken

INTRODUCTION

Salmonellosis is a broad term that refers to acute and chronic infections caused by
*Salmonella* in animals and humans and is a serious public health concern (1). *Salmonella*,
a gram-negative rod, is a major foodborne zoonotic pathogen with a wide host range (2).
*Salmonella enterica* serovar Enteritidis (*S. Enteritidis*) is a global concern, causing
recessive infections in adult chickens and bacterial excretion into the ex
environment through feces, resulting in challenging pathogen purification
systemic infections, and high mortality (3). In recent years, it has been
Enteritidis is gradually replacing *S. Typhimurium* as the most common
*Salmonella* serotype (4). Additionally, the widespread use of antibiotics has led to the
emergence of multiple antibiotic-resistant bacteria. Therefore, there is an urgent need for
an efficient vaccine to manage this major zoonotic infection (4). Owing to their ability to
stimulate the cellular and humoral adaptive immune responses, live *Salmonella* vaccines
are considered more effective against intestinal and systemic infections than inactivated
vaccines (5, 6).

Lipopolysaccharide (LPS), an important component of the outer membrane of gram-
negative rods, primarily influences the phenotype, virulence, and immune cross-reaction
of *Salmonella* strains (7, 8). LPS is primarily composed of O antigen, core
polysaccharides, and lipid A (9). The *rfa*, *rfb*, and *rfc* gene clusters are the main genes
involved in LPS biosynthesis (10). The *rfb* gene cluster regulates biosynthesis of the O
antigen chain, with *rfbG* encoding CDP-glucose 4,6-dehydratase (11). Jiao et
al. (12) showed that *rfbG* downregulation leads to LPS deficiency in *S. Enteritidis* which
reduced *S. Enteritidis* virulence in a mouse model, and that its gene locus can be used as a

redacted

2022-05-26 18:15:16

Line 61: What is a recessive infection?

check word choice.

foodborne

has led to the

79 marker. However, the effects of the *S. Enteritidis rfbG*-deficient strain in chicken remain
unknown.

This study aimed to determine whether attenuated levels of *S. Enteritidis rfbG*-deletion
mutants in a chicken model would protect poultry against virulent *S. Enteritidis*
challenge, while also allowing us to distinguish between the sera of vaccinated
infected chickens.

RESULTS

**Virulence of the vaccine candidates.** The virulence of the Z11 Δ *rfbG* and Z11 strains
(*rfbG*-deletion mutant strain and wild-type *S. Enteritidis* strain, respectively) was
evaluated in 7-day-old SPF chickens after intramuscular immunization. As shown in
Table 1, the LD₅₀ (50% lethal dose) of Z11 Δ *rfbG* was 1.9 \times 10⁹ CFU (colony-forming
unit), which was 61-fold higher than that of the virulent wild-type Z11 (3.1 \times 10⁷ CFU).
These results indicated that the virulence of Z11 Δ *rfbG* was attenuated compared to that
of the wild-type strain.

**Immune protective efficacy of Z11 Δ *rfbG* after bacterial challenge.** To evaluate the
protective efficacy of Z11 Δ *rfbG*, chickens were vaccinated intramuscularly with
Z11 Δ *rfbG* and challenged with the virulent wild-type strain Z11. The survival
percentages of these chickens are shown in Table 2. Following the challenge with Z11,
two chickens died in the 5 \times 10⁵ CFU-vaccinated group. Meanwhile, 4 out of 15 non-
vaccinated chickens died in the control group. However, none of the chickens in the
5 \times 10⁶ and 5 \times 10⁷ CFU groups died. As shown in Figure 1, there was a significant weight
loss in the non-vaccinated group compared to the vaccinated and control groups at 11

redacted

2022-05-26 18:16:05

Line 83: See comment for line 43 regarding importance of differentiation

102 days post-challenge (DPC). However, there was no significant difference between the
103 5×10^6 and 5×10^7 CFU-inoculated groups and the control group. Slight

diarrhea was observed following challenge in the vaccinated group compared to the
control group, whereas persistent weight loss, diarrhea, ruffled feathers,
were observed following challenge in the non-vaccinated group.

**Clearance of bacteria in the spleen and liver.** Liver and spleen samples from three
chickens in each group were aseptically collected 14 DPC to evaluate the bacterial
clearance in the internal organs of the chickens. The bacterial isolates were identified in
the liver and spleen of the non-vaccinated group but not in the liver or the spleen of the
vaccinated groups at 14 DPC. These results indicated that the vaccine promoted bacterial
clearance from the liver or spleen.

**Histological analysis after challenge with Z11.** Liver lesions were observed at 14
DPC and histological analysis of the liver was performed using Hematoxylin and eosin
(H&E) staining. As shown in Figure 2, no obvious lesions were detected in the livers of
the 5×10^6 and 5×10^7 CFU-vaccinated groups compared to those of the control group.
However, white nodules were observed in the liver of the non-vaccinated group (Figure
2A), and the liver cells of the 5×10^5 CFU-vaccinated group and the non-vaccinated group
had fatty changes which were more severe in the non-vaccinated group (Figure 2B).

**Humoral immune responses after immunization.** Serum *S. Enteritidis*-specific IgG
antibodies were evaluated in chickens following Z11 $\Delta rfbG$ immunization to determine
the efficacy of Z11 $\Delta rfbG$ in inducing humoral immune responses. As shown in Figure 3,
IgG antibodies

redacted
2022-05-26 18:18:47

Line 103: Please ensure all clinical symptoms noted are mentioned in methods section as symptoms to monitor

against *S. Enteritidis* in the Z11 Δ *rfbG* vaccinated group was detected and increased
dramatically 10 days after the second immunization. The 5×10^6 and 5×10^7 CFU-
vaccinated groups had significantly higher *S. Enteritidis*-specific IgG antibody titers than
the control group.

**The DIVA capability of the Z11 Δ *rfbG* vaccine.** The DIVA capability of the
Z11 Δ *rfbG* vaccine was evaluated using the slide agglutination test or the Biocheck
*Salmonella* group D antibody ELISA test to detect LPS-specific serum antibodies. The
slide agglutination test showed that the serum samples from chickens immunized with
Z11 Δ *rfbG* failed to agglutinate with the commercialized agglutination antigens on day 14
post-immunization (Figure 4). However, serum samples from chickens infected with the
virulent wild-type strain Z11 agglutinated with *S. Enteritidis* antigens (Figure 4).

**DISCUSSION**

*Salmonella enterica* causes high morbidity and death in humans, both in terms of the
number of infections and severity of the disease. In contrast to the *Salmonella enterica*
serovars with a limited host range, such as *S. Typhi*, *S. Dublin*, and *S. Gallinarum*, *S.*
*Enteritidis* has a large host range (13). Additionally, poultry and poultry-associated
products are among the most important vehicles for human *Salmonella* infections (14).
Vaccination in chickens is an important strategy currently used to reduce the levels of
*Salmonella* in poultry flocks (15). A live attenuated *Salmonella* vaccine must be
sufficiently attenuated, immunogenic, and protective. A variety of *Salmonella* mutant
strains have been suggested as candidate vaccines. In a previous study, we constructed a
mutant of the *S. Enteritidis* Z11 strain that lacks *rfbG* (16). The objective of the present

study was to evaluate the safety, protective efficacy, and DIVA capability of the *rfbG*-
deficient mutant *S. Enteritidis* Z11 strain as a live attenuated DIVA vaccine candidate.

Live attenuated *Salmonella* vaccines are avirulent. Some LPS-deficient mutants result
in attenuation of *Salmonella* (17). Jiao et al. (12) showed that the virulence of *rfbG*
mutant strains was attenuated in a mouse model, and that *rfbG* mutant strains are safe for
mammals. Similarly, this study showed that the LD₅₀ of Z11Δ*rfbG* was 61-fold higher
than that of wild-type Z11 in a chicken model, indicating that the virulence of Z11Δ*rfbG*
was significantly attenuated compared to that of virulent wild-type strain Z11 in
chickens. In contrast, *rfaH*, an essential gene for LPS biosynthesis, does not affect the
virulence of *S. Gallinarum* (18). Although the virulence of *S. Pullorum*
S06004Δ*spiC*Δ*rfaH* was significantly attenuated compared to that of wild-type S06004, it
was mainly the *spiC* deletion that exerted the attenuating effect (19).

The early immune response against *Salmonella* relies on the innate immunity within
the gut mucosa. In general, antibodies are necessary for resistance to systemic *Salmonella*
infections, and as the infection progresses, an effective immune response to *Salmonella*
relies on humoral immunity, which can provide effective protection to the host (20-21). A
previous study reported that a novel live *S. Enteritidis* vaccine strain, JOL919, induced
significant systemic immunoglobulin responses in chickens after inoculation (22).
Another study reported that IgG levels were significantly higher in chickens vaccinated
with the *cobS* and *cbiA* mutant *S. Gallinarum* strains than in the control group (23).
Similarly, in this study, all chickens immunized with 5×10^6 and 5×10^7 CFU Z11Δ*rfbG*
had significantly higher levels of *S. Enteritidis*-specific serum IgG than the non-

vaccinated group. These results demonstrated that the *rfbG* mutant can induce strong
humoral responses.

A standard vaccine not only induces a strong immune response but also has high
immune protection efficacy. In this study, the survival rate of the 5×10^7 and 5×10^6 CFU-
vaccinated groups was 100% after two intramuscular injections. Furthermore, chickens
immunized with 5×10^6 and 5×10^7 CFU of Z11 Δ *rfbG* showed no clinical symptoms.
Slight pathological changes were observed in the organs of 5×10^5 CFU-vaccinated
chickens, whereas more severe fatty changes were detected in unvaccinated chickens
after the challenge. *S. Enteritidis* colonize the gastrointestinal tract in avians, which is
frequently followed by an invasion of systemic areas, including the spleen and liver (24).
Braukmann et al. (25) showed that a live *Salmonella enterica* vaccine effectively
inhibited the invasion and colonization of the challenge strain. Furthermore, a study
reported that bacterial clearance occurred one week after *Salmonella* infections (26).
Sven et al. (27) demonstrated that a live *Salmonella Enteritidis* vaccine inhibits systemic
invasion after early infection with *S. Typhimurium* and *S. Infantis*. Our results showed
that intramuscular immunization in 7-day-old chickens with a mutant strain of *S.*
*Enteritidis rfbG* resulted in lower levels of the challenge strain and complete clearance of
bacteria in the liver and spleen, which is consistent with previous studies. Therefore, the
*rfbG* mutant showed a strong protective efficacy.

The DIVA vaccine, an identifiable vaccine, can be used to distinguish immunized
animals from infected animals. Most of the attenuated mutant strains used in DIVA
vaccines lack specific antigens (such as LPS and neuraminidase); therefore, LPS is a
suitable target for the construction of the DIVA vaccine (28). We have previously shown

that *rfbG* plays a role in LPS synthesis (12). In the present study, the Z11 Δ *rfbG*
vaccinated group exhibited a negative serological test result, while the virulent wild-type
Z11 infection group showed a strong positive result for the serological test. These results
indicated that the *rfbG* mutant allows for the differentiation between infected animals and
vaccinated animals. Therefore, it may be used in combination with herd-level *Salmonella*
surveillance.

In conclusion, this study demonstrated that *S. Enteritidis* Z11 Δ *rfbG* has the potential to
be a safe, immunogenic, and DIVA vaccine candidate for the control of *Salmonella*
infections. Attenuated *S. Enteritidis* Z11 Δ *rfbG* elicited a strong humoral immune
response and provided effective protection in chickens. In addition, vaccination with
Z11 Δ *rfbG* did not affect our ability to distinguish between infected and vaccinated
chickens by serological tests.

MATERIALS AND METHODS

**Experimental animals.** Healthy SPF chickens (7-day-old) were obtained from
Zhejiang Lihua Agricultural Technology Co., Ltd, China. All animal experiments were
approved by the Animal Welfare and Ethics Committees of Yangzhou University and
complied with the guidelines of the Institutional Administrative Committee and Ethics
Committee of Laboratory Animals (IACUC license number: SYXK [Su] 2016–0020).

**Bacterial strains.** Virulent wild-type *S. Enteritidis* strain Z11 was a clinical isolate
obtained from *S. Enteritidis*-infected chickens and stored in our laboratory. The *rfbG*-
deletion mutant strain, Z11 Δ *rfbG*, was contrasted using a homologous recombination
technique mediated by a suicide plasmid, as previously described (16). Briefly, upstream

and downstream fragments of the *rfbG* gene were amplified using PCR. The pDM4
plasmid was digested using a restriction endonuclease *Xba* I (TaKaRa). The purified
plasmid and the upstream and downstream fragments were fused using the ClonExpress
MultiS One Step Cloning Kit (Vazyme Biotechnology Co., Ltd., Nanjing, Jiangsu,
China). Recombinant plasmids were transferred into X7213 cells and sequenced. Single
crossover mutants were obtained by conjugal transfer of recombinant suicide plasmids
into the Z11 strain. The *rfbG*-deletion mutant was screened on 15% sucrose Luria-Bertani
(LB) plates. The *S. Enteritidis* Z11 Δ *rfbG* strain was used as a DIVA vaccine candidate in
this study.

**Assessment of bacterial virulence.** The virulence of *S. Enteritidis* Z11 and Z11 Δ *rfbG*
vaccines was evaluated in chickens by determining the 50% lethal dose (LD₅₀). Sixty-six
7-day-old SPF chickens were used in this study. Thirty chickens in the wild-type group
were randomly assigned to five groups (n=6). Each group was injected intramuscularly
with a 10-fold dilution (1×10^5 to 1×10^9 CFU) of Z11. Thirty chickens in the deletion
mutant group were randomly assigned to one of five groups (n=6). Each group was
injected intramuscularly with a 10-fold dilution (2×10^6 – 2×10^{10} CFU) of Z11 Δ *rfbG*. Six
chickens were inoculated with 100 μ L of phosphate-buffered saline (PBS) via the same
route as the control group. Chicken death was monitored daily for 14 days post-infection.
LD₅₀ was calculated using the Reed–Muench method (29).

**Immune protection assessment.** Seventy-five 7-day-old SPF chickens were
randomly assigned to three groups, namely the control (no vaccine, no challenge) (n=15),
bacterial challenge without immunization (n=15), and live vaccine groups with the
bacterial challenge (n=45). SPF chickens in the vaccine groups were randomly assigned

to three groups (n=15). Each group was administered 100 μ L of diluted suspensions of
Z11 Δ *rfbG* containing 5×10^8 , 5×10^7 , or 5×10^6 CFU/mL in PBS by intramuscular
injection. Control chickens received 100 μ L PBS via the same route. After the first
immunization, the vaccine groups were administered a booster dose at 17 d of age. These
chickens, as well as those in the unvaccinated group (28-day-old), were challenged
intramuscularly with 7×10^8 CFU of the Z11 strain 11 days after the second
immunization. Deaths and clinical symptoms were recorded daily for 14 days after the
challenge.

**Bacterial clearance assay.** Bacterial clearance in the internal organs of the chickens
was evaluated. Liver and spleen samples from three chickens in each group were
aseptically collected at 14 DPC. The samples were then weighed and homogenized in 1
249 mL of PBS. Homogenates were 10-fold serially diluted and subsequently inoculated onto
250 LB agar plates at 37 °C for 12–16 h. Bacterial colonies were calculated as log₁₀ CFU/g.

**Histological analysis.** Sections of the spleen and liver were collected from chickens at
14 DPC, and tissue samples were fixed in 10% neutral-buffered formalin. Paraffin-
embedded sections were stained with H&E stain (30) and were observed at 100 \times and
400 \times magnifications using an optical microscope.

**Serum IgG test.** Humoral immune responses were evaluated by determining the *S.*
Enteritidis-specific IgG antibody levels using enzyme-linked immunosorbent assay
(ELISA), as previously described (31), using Z11 Δ *rfbG* as the coating antigen. Serum
samples were collected from chickens in each group on day 10 after the second
immunization and then serially diluted to be used as the primary antibody. The secondary
antibody used was horseradish peroxidase (HRP)-conjugated rabbit anti-chicken IgG

(1:10,000 dilution; Sigma-Aldrich, St. Louis, MO, USA). HRP activity was determined
using a 3,30,5,50-tetramethylbenzidine substrate solution (Solarbio, Beijing, China), and
the optical density (OD₄₅₀) value was determined using an ELISA reader (BioTek,
Winooski, VT, USA).

**DIVA capability assessment for the Z11ΔrfbG vaccine.** The DIVA capability of the
Z11ΔrfbG strain was evaluated using the serological method to detect LPS-specific
serum antibodies by a slide agglutination test or a commercial ELISA kit. Fifteen
chickens were randomly divided into 3 groups (n=5). Cells were infected with Z11ΔrfbG,
Z11, or PBS. Serum was collected 14 days later and used to detect LPS antibodies. The
slide agglutination test was performed using commercialized agglutination antigens
obtained from Zhonghai Biotech Co., Ltd. (Beijing, China) according to the
manufacturer's instructions. ELISA was performed using the *Salmonella* group D
antibody test kit (BioCheck, Inc., San Francisco, CA, USA) according to the
manufacturer's instructions (19).

**Statistical analysis** GraphPad Prism 5 software (San Diego, CA, U.S.)
data analysis. Data are presented as mean ± standard error (SEM). Statistical
was set as $P < 0.05$ (*), < 0.01 (**), or 0.001 (***)

redacted
2022-05-26 19:18:10

Line 275: Each statistical test done
needs
to be describe in the methods section
and mentioned in each figure legend.
No statistical tests are mentioned in
this manuscript.

279 ACKNOWLEDGMENTS

[revised manuscript text omitted]

- 22. Nandre RM, Matsuda K, Chaudhari AA, Kim B, Lee JH. 2011. A genetically
engineered derivative of *Salmonella* Enteritidis as a novel live vaccine candidate for
salmonellosis in chickens. Res Vet Sci 93:596-603.
- 23. Penha Filho RAC, Diaz SJA, Medina TdS, Chang Y-F, da Silva JS, Berchieri A.
2016. Evaluation of protective immune response against fowl typhoid in chickens
vaccinated with the attenuated strain *Salmonella* Gallinarum $\Delta cobS\Delta cbiA$. Res Vet
Sci 107:220-227. DOI: 10.1016/j.rvsc.2011.11.005.
- 24. Wigley P. 2017. *Salmonella* enterica serovar Gallinarum: addressing fundamental
questions in bacteriology sixty years on from the 9R vaccine. Avian Pathol 46:119-
124. DOI: 10.1080/03079457.2016.1240866.
- 25. Braukmann M, Barrow PA, Berndt A, Methner U. 2016. Combination of competitive
exclusion and immunization with a live *Salmonella* vaccine in newly hatched
chickens: Immunological and microbiological effects. Res Vet Sci 107:34-41. DOI:
10.1016/j.rvsc.2016.05.001.
- 26. Takaya A, Yamamoto T, Tokoyoda K. 2019. Humoral Immunity vs. *Salmonella*. Front
Immunol 10:3155-3161. DOI: 10.3389/fimmu.2019.03155.
- 27. Sven Springer, Tobias Theuß, Imre Toth, Zsuzsanna Szogyenyi. 2021. Invasion
inhibition effects and immunogenicity after vaccination of SPF chicks with a
*Salmonella* Enteritidis live vaccine. Tierarztl Prax Ausg G Grosstiere Nutztiere
49:249-255. DOI: 10.1055/a-1520-1369.
- 28. Maas A, Meens J, Baltes N, Hennig-Pauka I, Gerlach G-F. 2006. Development of a
DIVA subunit vaccine against *Actinobacillus pleuropneumoniae* infection. Vaccine
24:7226-7237. DOI: 10.1016/j.vaccine.2006.06.047.

- 29. Reed LJ. 1938. A simple method of estimating fifty percent endpoints. Am J Hyg
27:493-497. DOI: 10.1093/oxfordjournals.aje.a118408.
- 30. Cardiff RD, Miller CH, Munn RJ. 2014. Manual hematoxylin and eosin staining of
mouse tissue sections. Cold Spring Harb Protoc. 2014:655-658. DOI:
10.1101/pdb.prot073411.
- 31. Wang Y, Huang C, Tang J, Liu G, Hu M, Kang X, Zhang J, Zhang Y, Pan Z, Jiao Xa.
2021. *Salmonella* Pullorum *spiC* mutant is a desirable LASV candidate with proper
virulence, high immune protection, and easy-to-use oral administration. Vaccine
39:1383-1391. DOI: 10.1016/j.vaccine.2021.01.059.

**TABLES AND FIGURES**

**Table 1.** The 50% lethal dose (LD₅₀) of the *S. Enteritidis* Z11 and Z11Δ*rfbG* strains in
 chickens.

Strains	Challenge dose (CFU)	Number of deaths/Total number of chickens	LD ₅₀ (CFU)
Z11	1×10 ⁹	6/6	3.1×10 ⁷
	1×10 ⁸	5/6	
	1×10 ⁷	1/6	
	1×10 ⁶	0/6	
	1×10 ⁵	0/6	
Z11Δ rfbG	2×10 ¹⁰	5/6	1.9×10 ⁹
	2×10 ⁹	4/6	
	2×10 ⁸	0/6	
	2×10 ⁷	0/6	
	2×10 ⁶	0/6	
Blank	PBS	0/6	/

**Table 2.** The protective efficacy of intramuscular Z11 Δ *rfbG* vaccination.

Vaccination			Challenge				Survivors/ Total	Survival rate (%)
Strain	Route	Dose (CFU)	Number	Strain	Route	Dose (CFU)		
		5×10^5					13/15	86.7
Z11 Δ rfbG		5×10^6					15/15	100
	Intramuscularly	5×10^7	15	Z11	Intramuscularly	7×10^8	15/15	100
PBS		-					11/15	73.3
PBS		-					15/15	100

**Figure 1.** The bodyweight of chickens after the challenge. Chickens of vaccinated and
non-vaccinated groups were intramuscularly challenged with 7×10^8 colony-forming
units (CFU) of the virulent wild-type strain (Z11), and the control group received 100 μ L
of PBS. The bodyweights of these chickens were recorded 11 days post-challenge (DPC).
*, $P < 0.05$, **, $P < 0.01$, and ***, $P < 0.001$ compared with the bodyweight of control
group chickens. Data are presented as mean \pm SEM.

**Figure 2.** A diagram showing the pathological anatomy (A) and histological analysis (B)
of the liver after the bacterial challenge. (A) The liver pathological anatomy diagram was
observed 14 days post-challenge (DPC). The arrows indicate the lesions. (B) The
histopathological changes in the livers of chickens were examined by H&E staining 14
DPC. The results were observed at $100 \times$ and $400 \times$ magnification using an optical
microscope. Arrows in the liver sections represent the fatty changes. The circle in the
liver section indicates the necrotic foci.

**Figure 3.** Determination of serum IgG levels. The enzyme-linked immunosorbent assay
was used to identify *S. Enteritidis*-specific IgG antibody titers in the serum of chickens
from each group 10 days following the second inoculation. *, $P < 0.05$, and ***, $P <$
0.001 compared with the control group. Data are presented as mean \pm SEM.

**Figure 4.** The DIVA capability of $Z11\Delta rfbG$. The serum was collected from chickens
infected with $Z11\Delta rfbG$, Z11, or PBS for 14 days and used for the detection of LPS
antibodies. Agglutination assay was performed using commercialized agglutination
antigens.

**Figure 1.**

**Figure 2.**

**Figure 3.**

**Figure 4.**

Table 1. The LD₅₀ of the *S. Enteritidis* Z11 and Z11Δ*rfbG* in chickens.

Strains	Challenge dose (CFU)	Number of deaths/Total number of chickens	LD ₅₀ (CFU)
Z11	1×10 ⁹	6/6	3.1×10 ⁷
	1×10 ⁸	5/6	
	1×10 ⁷	1/6	
	1×10 ⁶	0/6	
	1×10 ⁵	0/6	
Z11Δ rfbG	2×10 ¹⁰	5/6	1.9×10 ⁹
	2×10 ⁹	4/6	
	2×10 ⁸	0/6	
	2×10 ⁷	0/6	
	2×10 ⁶	0/6	
Blank	PBS	0/6	/

Table 2. The protective efficacy of the *Z11Δ*rfbG** after intramuscular vaccination.

Vaccination			Challenge				Survivors/ Total	Survival rate (%)
Strain	Route	Dose (CFU)	Number	Strain	Route	Dose (CFU)		
Z11ΔrfbG		5×10^5					13/15	86.7
		5×10^6					15/15	100
	Intramuscularly	5×10^7	15	Z11	Intramuscularly	7×10^8	15/15	100
PBS		-					11/15	73.3
PBS		-					15/15	100

Figure 1. The bodyweight of chickens after the challenge. Chickens of vaccinated and non-vaccinated groups were intramuscularly challenged with 7×10^8 colony-forming units (CFU) of the virulent wild-type strain (Z11), and the control group received 100 μ L of PBS. The bodyweights of these chickens were recorded 11 days post-challenge (DPC). *, $P < 0.05$, **, $P < 0.01$, and ***, $P < 0.001$ compared with the bodyweight of control group chickens. Data are presented as mean \pm SEM.

Figure 2. A diagram showing the pathological anatomy (A) and histological analysis (B) of the liver after the bacterial challenge. (A) The liver pathological anatomy diagram was observed 14 days post-challenge (DPC). The arrows indicate the lesions. (B) The histopathological changes in the livers of chickens were examined by H&E staining 14 DPC. The results were observed at $100 \times$ and $400 \times$ magnification using an optical microscope. Arrows in the liver sections represent the fatty changes. The circle in the liver section indicates the necrotic foci.

Figure 3. Determination of serum IgG levels. The enzyme-linked immunosorbent assay was used to identify *S. Enteritidis*-specific IgG antibody titers in the serum of chickens from each group 10 days following the second inoculation. *, $P < 0.05$, and ***, $P < 0.001$ compared with the control group. Data are presented as mean \pm SEM.

Figure 4. The DIVA capability of *Z11ΔrfbG*. The serum was collected from chickens infected with *Z11ΔrfbG*, *Z11*, or PBS for 14 days and used for the detection of LPS antibodies. Agglutination assay was performed using commercialized agglutination antigens.

September 5, 2022

Dear Editor:

We have revised the manuscript of Spectrum01574-22 (Evaluation of the protective immune response induced by an *rfbG*-deficient *Salmonella* Enteritidis strain as a live attenuated DIVA vaccine in chickens) and made changes as suggested by the reviewers and the editor. All the reviewers' and editorial comments have been accepted.

Enclosure:

1. The revised manuscript and all changes of the manuscript marked in yellow color;
2. The letter detailed our responses to all the comments passed by the reviewers and the editor.

Thank you for your attention!

Yours sincerely

Hongqin Song

Yangzhou University

The letter accepting changes to the reviewer 1

Dear reviewer:

Thank you very much for your valuable comments on our manuscript. We have revised the manuscript of Spectrum01574-22 (Evaluation of the protective immune response induced by an *rfbG*-deficient *Salmonella* Enteritidis strain as a live attenuated DIVA vaccine in chickens) and made changes as you suggested.

Reviewer 1: Overall this is an interesting study with well supported conclusions. Authors should add all statistical test methods used in the paper in the methods section and figure legends. Additionally, authors can elaborate on the importance of being able to differentiate between vaccinated and infected chickens using sera.

1. Authors should add all statistical test methods used in the paper in the methods section and figure legends.

Answer: Thanks for your kind advice. In the methods section (line 310) and in figure legends, we added the statistical test methods used in the results (one-way ANOVA and Bonferroni's multiple comparison test).

2. Additionally, authors can elaborate on the importance of being able to differentiate between vaccinated and infected chickens using sera.

Answer: Thank you for your kind suggestion. Because *Salmonella* vaccination may interfere with existing serologic monitoring, the development of a vaccine that distinguishes between immunized and infected animals is necessary. In line 204, we describe the importance of distinguishing between immunized and infected animals.

The letter accepting changes to the reviewer 2

Dear reviewer:

Thank you very much for your valuable comments on our manuscript. We have revised the manuscript of Spectrum01574-22 (Evaluation of the protective immune response induced by an *rfbG*-deficient *Salmonella* Enteritidis strain as a live attenuated DIVA vaccine in chickens) and made changes as you suggested.

Reviewer 2:

Evaluation of the protective immune response induced by an *rfbG*-deficient *Salmonella* Enteritidis strain as a live attenuated DIVA vaccine in chickens by Wang et al.

The authors claim to have constructed a *rfbG* mutant of *Salmonella enterica* serovar Enteritidis (SE) that would be suitable as a live attenuated DIVA vaccine strain with protective properties against SE infections.

The topic of SE as a pathogen for poultry, but more importantly as a zoonotic pathogenic bacterium affecting humans, thus being a public health concern, is timely.

This reviewer can imagine how much effort was put into the study. Nonetheless, the manuscript appears at this state somewhat preliminary as all sections of the manuscript need to be more elaborated. Many sections appear rather cryptic and are lacking details that are essential for the understanding of the executed experiments.

Grammar and style should be carefully checked.

Answer: Thank you for your kind advice. We described the experimental methods and results in detail in the manuscript, such as a detailed description of the experimental

groups (line 254). In view of the grammar problem, we also had a professional embellishment company (Editage, <https://www.editage.cn>) to polish the manuscript and revise it.

A very important aspect, although mentioned in the discussion section, is totally excluded from their own study: the colonization of the gastrointestinal tract of the chickens after oral uptake/inoculation of SE. This plays an important role in the natural environment as well as in the small scale and largescale industrial settings of poultry production. Many issues of food poisoning through SE occur when contaminated poultry carcasses are processed. Protection of poultry against systemic infection is important, indeed. But what about protection against gastrointestinal colonization and shedding of SE into the environment? The main aspect still is the problem that this zoonotic pathogen can easily be transferred to humans via poultry products and hens eggs.

Answer: Thank you for your valuable suggestions for us. Therefore, based on your suggestion, we examined the colonization of bacteria in the liver, spleen, and cecum after challenge by a repeated bacterial colonization assay, and the results showed that the bacterial load in the vaccinated group was significantly reduced in the liver, spleen, and cecum compared with the non-vaccinated group, indicating that the *rfbG*-deficient strain accelerated the clearance of bacteria in the liver, spleen, and cecum, thus reducing the colonization.

Your question about how to prevent the transmission of *Salmonella* Enteritidis to humans through poultry products and eggs is very important to us, but since our

experimental design was developed based on published studies (1, 2), our study is still in the preliminary stage of vaccine protection. We will further investigate at a later stage, based on your suggestions, whether *rfbG*-deficient strains can reduce the transmission of *Salmonella enterica* pathogens in poultry products and eggs.

Specific comments:

1. It is rather safe to say that reversion of the *rfbG* mutation can be excluded. But what about the possibility of recombination with DNA fragments from other *Salmonella* strains that could potentially be present in the same bird at the time of vaccination? Please discuss.

Answer: Thank you for your suggestion. All the chickens we used in the experiment were SPF chickens, and before we started the experiment, we would strictly sterilize the isolator to ensure that it was completely sterile, and we also set up a control group to eliminate the contamination of other bacteria.

2. Abstract; first sentence: other serovars can cause salmonellosis as well. Please re-phrase.

Answer: Thank you very much for your comments. We are very sorry for this loose remark. We have rewritten the original sentence in line 26 as "*Salmonella enterica* serovar Enteritidis (*S. Enteritidis*), one of the zoonotic pathogens, not only results in significant financial losses for the global poultry industry, but it also has the potential to spread to humans through poultry and poultry products".

3. Line 34-39: Re-phrase, please. A lethal challenge is per definition deadly. How can it be that birds do not show any signs of disease after a lethal challenge?

Answer: Thank you for your kind advice. We have rewritten the original sentence in line 35 as "After challenge, the non-vaccinated group showed serious clinical symptoms (diarrhea, decreased appetite, depression, weight loss), pathological changes (white nodules in the liver and fatty lesions in liver cells) and death. In contrast, there were no clinical symptoms, pathological changes, or death in the 5×10^6 and 5×10^7 colony-forming unit (CFU)-vaccinated groups".

4. Line 42: Find another word for "transformed".

Answer: Thank you for your suggestion. We have rewritten the original sentence in line 44 as "Overall, our findings demonstrate the viability of the *rfbG* mutant as a live attenuated chicken vaccine that can discriminate between animals that have been immunized and those that have been infected."

5. Line 59 and throughout the document: "Gram" starts always with a capital "G".

Answer: Considering to your suggestion, we changed it to "Gram-negative".

6. Complete results section: Please provide sufficient details in all sub-sections.

Answer: Thank you for your kind advice. We have described the results in detail, such as the results of bacterial clearance (line 105) and changes in body weight (line 124) after challenge.

7. Line 119 and later in manuscript: what are "fatty changes"? Please explain.

Answer: Under normal circumstances, in addition to fat cells, other cells generally do not see or only a small number of lipid droplets, such as the emergence of lipid droplets in these cells or a significant increase in lipid droplets, called steatosis.

8. Line 213: Do the authors mean "constructed" rather than "contrasted"?

Answer: We are very sorry for our incorrect writing. We have changed "contrasted" to "constructed" in line 231.

9. Line 221 -223: How was the *rfbG* mutation confirmed? By sequencing?

Answer: We verified whether the *rfbG* was deleted successfully by PCR. The detailed process is as follows: the outer primers and inner primers of *rfbG* were designed respectively, and the deletion of *rfbG* was determined by amplifying the sequences of outer primers and inner primers. Using the wild-type strain Z11 as the negative control, if the sequence amplified by the outer primers was smaller than that of the negative control and the size of the different band met the size of the *rfbG* band but the inner primers could not amplify the sequence, the *rfbG* deletion was successful. We have added a method to verify the deletion strain in line 240.

10. Lines 246 ff: Which volume of the samples was plated onto LB plates? What is the detection limit? Samples should have undergone an enrichment procedure in addition to "direct plating" How would LB medium discriminate against bacterial strains other than the challenge strain?

Answer: 1. 100 μ L of tissue slurry was added dropwise to each LB plate. 2. We carried out bacteriological analysis according to the description of Barrow et al. (3). The steps of the bacterial colonization test are as follows: aseptically collect the organ tissues, place them in homogenized tubes containing sterile PBS, grind them to obtain tissue slurry, dilute the tissue slurry gradient 10 times, select the appropriate dilution gradient for plate coating according to the specific situation, incubate them at 37°C for 12-16 hours, and calculate the bacteria load of the organ the next day based on the

bacterial count results and the mass of the collected organ tissues. 3. PCR was used to distinguish between wild strains and deletion strains. In line 105 of the manuscript, we mention that before challenge (7 and 10 days after the second immunization), no bacteria were isolated from the liver, spleen, and cecum of the vaccinated group and the non-vaccinated group, indicating that there were no *rfbG* deletion strains in the vaccinated group and the non-vaccinated group. After challenge, we use PCR to verify the bacteria on the LB plate. The results show that the bacteria on the LB plate are wild strains. (For specific methods of PCR, see question 9)

11. Line 253: Please explain H&E stain.

Answer: Hematoxylin and eosin (H&E) stain is one of the most basic and widely used techniques in histology and pathology teaching and research. Hematoxylin staining solution is alkaline and can stain the basophilic structures of tissues (such as ribosomes, nuclei and ribonucleic acid in cytoplasm) into blue-purple; eosin is an acidic dye and can stain the eosinophilic structures of tissues (such as intracellular and intercellular proteins) into pink, so that the morphology of the whole cellular tissue is clearly visible. The tissue sectioning method is a very common test method in teaching, research, and pathology testing, and H&E stain is the most common staining method used in the process of making sections.

12. Table 2: Please elaborate what is the difference between the two PBS groups?

Answer: One group of PBS was not inoculated with the *rfbG* deletion strain but was challenged. We named it the non-vaccinated group in line 255. The other group of PBS was not inoculated with the *rfbG* deletion strain and was not challenged. We

named it the control group in line 254.

13. Table 1: The respective experiment should be executed in duplicate and repeated with a larger number of birds. Were the birds commingled or housed in different rooms (potential "pen effect")?

Answer: According to your suggestion, we expanded the number of samples and re-carried out the experiment. The experimental method and results have been modified in this manuscript. The results showed that there was no cross-contamination between different groups of chickens raised in different isolators and a control group.

Reference

1. Penha Filho RA, de Paiva JB, Arguello YM, et al. 2009. Efficacy of several vaccination programmes in commercial layer and broiler breeder hens against experimental challenge with *Salmonella enterica* serovar Enteritidis. Avian Pathol 383:67-375. DOI: 10.1080/03079450903183645.
2. Guo Y, Xu Y, Kang X, et al. 2019. Immunogenic potential and protective efficacy of a *sptP* deletion mutant of *Salmonella* Enteritidis as a live vaccine for chickens against a lethal challenge. Int J Med Microbiol 309:151337. DOI: 10.1016/j.ijmm.2019.151337.
3. Barrow PA, Lovell MA. 1991. Experimental infection of egg-laying hens with *Salmonella enteritidis* phage type 4. Avian Pathol 20:335-348. DOI: 10.1080/03079459108418769.

October 22, 2022

Dr. Hongqin Song
Yangzhou University
Yangzhou
China

Re: Spectrum01574-22R1 (Evaluation of the protective immune response induced by an *rfbG*-deficient *Salmonella* Enteritidis strain as a live attenuated DIVA vaccine in chickens)

Dear Dr. Hongqin Song:

Your manuscript has been accepted, and I am forwarding it to the ASM Journals Department for publication. You will be notified when your proofs are ready to be viewed.

Sincerely,

Mariola Edelman
Editor, Microbiology Spectrum